# Air-Coupled Ultrasonic Probe Integrity Test Using a Focused Transducer with Similar Frequency and Limited Aperture for Contrast Enhancement

**DOI:** 10.3390/s20247196

**Published:** 2020-12-16

**Authors:** Linas Svilainis, Andrius Chaziachmetovas, Darius Kybartas, Tomas Gomez Alvarez-Arenas

**Affiliations:** 1Department of Electronics Engineering, Kaunas University of Technology, LT-51368 Kaunas, Lithuania; andrius.chaziachmetovas@ktu.lt (A.C.); darius.kybartas@ktu.lt (D.K.); 2Instituto de Tecnologias Físicas y de la Información, CSIC, 28006 Madrid, Spain; t.gomez@csic.es

**Keywords:** ultrasound transducer, air-coupled ultrasound, beam sidelobes, probe characterization, focused transducer

## Abstract

Air-coupled ultrasonic probes require a special design approach and handling due to the significant mismatch to the air. Outer matching layers have to be soft so can be easily damaged and excitation voltages might cause the degradation of electrodes or bonding between the layers. Integrity inspection is desired during design, manufacturing, and exploitation. Spatial distribution of a transduction efficiency over piezoelement surface is proposed as a convenient means for the air-coupled probe integrity inspection. Focused transducer of similar center frequency is used to scan the surface of the inspected probe. However, such approach creates a challenge, i.e., area of the scanning beam is much smaller than the total receiving area of the inspected probe, therefore, contrast and imaging resolution are significantly degraded. Masking aperture made from cardboard and felt, placed at the focal point was proposed as solution. Far-range sidelobes were suppressed down to the noise floor (−50 dB) and the near-range sidelobes were reduced down to −17 dB. The proposed modification allows to use a similar frequency focused transducer. Probe integrity inspection can be carried out at significantly enhanced contrast and lateral resolution. Natural and artificial defects can be detected by the use of the proposed method.

## 1. Introduction

The use of air-coupled ultrasound offers a convenient means in the nondestructive testing and measurements in the case when the coupling (contact, water immersion, or jet) is not applicable [1,2,3]. However, the design of the air-coupled ultrasonic probes is quite demanding due to the significant impedance mismatch between the piezoelectric element and the air. Impedance matching is usually implemented as a stack of matching layers [4,5,6]. Design and fabrication of the stack of matching layers require testing of the intermediate steps of the matching layers’ layout. It is also important to keep the beam homogeneous in near-zone applications [7]. Deviation of the probe’s beam pattern can cause incorrect inspection or diagnostics results [1,8,9]. Conformity of the focal spot is important in many applications [10,11,12,13]. Probe’s integrity may be lost during exploitation because outer matching layers are soft and they can be easily damaged [14,15]. High excitation voltages are usually used, causing the electrode detachment or chipping. Therefore, even the finished transducer requires inspection for the delaminations between the layers, matching layer dents, or degradation and electrode circuit continuity [8,16,17]. Conventional techniques for the integrity inspection use the sensitivity check of the probe [18], impedance measurement [19,20], or directivity testing [21]. Sensitivity measurement is a convenient means for probes comparison, evaluation of degradation, or design, assembly steps control [9,11,16]. Electrical impedance of the probe is an indication of electromechanical coupling efficiency [20]. It is a simple technique, allowing for quick estimation of resonant frequencies and electrical matching efficiency [22]. Probe’s directivity is the most important property [16]. It is usually measured in pulse-echo mode or probe’s field is investigated using hydrophone or microphone [23,24,25]. Yet, all the aforementioned approaches use the properties that are integral so cannot account nor locate local defects. Local properties evaluation was proposed in [23,24], where acoustic field measurement results are backward reradiated by reversibility of the wave process. Another approach, proposed in [25] measured the velocity distribution on the transducer surface to calculate the acoustic field generated by the transducer. Spatial distribution of the transduction efficiency over piezoelement surface could be a convenient means for the integrity inspection, allowing to locate local defects or damages as well as any sensitivity loss. Such technique was reported in [26]. It uses a focused beam scan over the inspected probe’s surface. Response to this beam at the tested probe’s output is used as local sensitivity measure. However, it is dedicated for water, and also a higher frequency (20 MHz) focused transducer is required to test a lower frequency (2–5 MHz) probe. Such approach cannot be used for the air-coupled probe because (i) quarter wavelength relationship of matching layers may be lost [4,5], (ii) high-frequency ultrasound has large attenuation in air [27,28,29], and (iii) such focused transducers are rare [30,31]. This paper presents an advanced approach, i.e., application is expanded to the air and, to permit that, the frequency range of the probing transducer and the inspected probe is close. Such application can be compared to probes’ surface vibrometry [6], only in reverse, transduction in reception is investigated. In contrast, proposed technique does not require reflecting coating (which is not applicable on air-matched layers) nor expensive and complex equipment. Another novelty of the proposed approach is the use of an aperture to mask the beam sidelobes of the probing transducer. Mask was realized as simple, low-cost hood mounted on the probing transducer. 

## 2. Materials and Methods

The idea was to scan the surface of the air-coupled unfocused (flat beam) probe with the focused beam of another transducer having a similar center frequency. If the spot of the focused beam is small enough, only a small fraction of the tested probe’s piezosensitive surface is excited. The amplitude of the received signal in such case is a representative of the sensitivity at this spot. Scanning the whole probe’s surface will produce a sensitivity map of the probe under inspection. 

### 2.1. Measurement Setup

Measurement setup is presented in Figure 1. The probe under investigation was fixed face-up while focused transducer was mounted on *x*-*y*-*z* positioning stage. While *x* and *y* coordinates were used for scanning, *z* coordinate positioning was used for focal distance matching on probe’s surface.

A narrowband (10% at −6 dB), 0.9 MHz, 15 mm diameter focused transducer manufactured at Ultrasound Research Institute, Kaunas University of Technology (Kaunas, Lithuania) [32] was used for transmission. The focal spot (29 mm distance, 1.5 mm in diameter) was placed on the tested probe’s surface. Focusing was achieved by the concave (29 mm radius) surface, formed by the manually produced composite piezoelement (note the small rectangles in Figure 2). This transducer contained two separate sections, because it was intended for the dual crystal use. Both sections were used in parallel for transmission in the current investigation. The width of the slit between the sections was 0.4 mm. The slit between the sections was oriented along the *y* scanning axis.

Directivity profiles of the transducer were measured in several modes. Figure 3 and Figure 4 show the unmodified transducer directivity profile. Figure 3 was obtained in reflection (3 mm ball as point reflector) mode. Transducer was excited by half bridge topology pulser SE-TX01-01 (Kaunas University of Technology, Kaunas, Lithuania) [33] using unipolar 200 V rectangular spread spectrum signal (0.7–1.1 MHz range, 40 µs chirp). Variable gain preamplifier PA1008 (AUT Solutions, Fulshear, TX, USA) was used for reception.

The beam width (at −6 dB) is 0.77 mm along *x*-axis and 0.97 mm along *y* axis in reflection mode. The slit (see Figure 2), separating the transducer sections, created the near-range sidelobes along x-direction. In reflection mode, near-range sidelobes are, approximately, at −10 dB. Uneven transmission from the surface, caused by the granular structure of the composite piezoelement [21], produced the far-range sidelobes. The far-range sidelobes are low in the reflection mode (−31 dB in few places, but mainly −45 dB and below). Such level of the sidelobe is acceptable for majority of imaging applications. However, here this transducer is used in transmission mode, and therefore, its directivity pattern is more complex (Figure 4). This field was measured using a 0.5 mm hydrophone (SN2331, Precision Acoustics Ltd., Dorset, UK). Hydrophone was used in a quite unusual way (air reception), therefore, the signal received was very weak, i.e., 0.3 Vpp after 58 dB amplification (8 dB Precision Acoustics, Dorset, UK, preamplifier PA15053 followed by 50 dB variable gain preamplifier PA1008 AUT Solutions, Fulshear, TX, USA). Averaging of 100 waveforms was used to obtain the acceptable signal to noise ratio (SNR). In addition to low sensitivity, operation in air makes impossible to use the hydrophone calibration chart and the frequency band is surely affected, yet obtained directivity patterns are correct.

The beam width (−3 dB) in transmission mode is 0.89 mm along *x*-axis and 1.0 mm along *y* axis. The slit-induced near-range sidelobes along x-direction are at −6 dB. Meanwhile, the far-range sidelobes are much higher, up to −18 dB. The level of the far-range sidelobe’s is essential in the envisaged application, i.e., area of the tested probe is higher than the area covered by the mainlobe. The mainlobe area is 0.7 mm^2^ according to measurement results, while the area of the inspected flat-beam probe is 314 mm^2^ (20 mm diameter), which is almost 500 times larger. This produces 53 dB gain for the sidelobes, providing that even the low-level, uncorrelated in phase, sidelobes might create the significant output at the receiving probe. As a result, the image contrast is reduced and small defects are masked.

Therefore, it was proposed to use an aperture, which would mechanically gate-out the sidelobes. This aperture cannot be placed at the focal distance, because inspected probe has a protective rim so piezoelement surface is 1.1 mm deeper than the front-surface. The mask (refer Figure 5) was constructed using 24 mm diameter and 0.5 mm thick cardboard and 1.5 mm felt textile as absorber.

Main contribution in the sidelobes cancellation comes from the cardboard aperture (d = 4 mm) placed in front of the mask. The felt layer is placed outside to provide some apodization on the cardboard aperture (d = 2.7 mm vs. 4 mm for cardboard). It cancels the reflections and reduces the diffraction effects of the cardboard aperture. Alignment of the beam and the aperture was done in a few steps: (i) the beam was positioned on ball-reflector (maximum reflected signal) without mask; (ii) then, the mask was mounted so that the reflector was inside the aperture hole; (iii) focused transducer was lowered by 3 mm, mask slid on the transducer, ensuring both beam centering and focal point protrusion by 2 mm outside the aperture. Refer Figure 6 for the masking aperture influence on the transmission field measured by the same 0.5 mm hydrophone.

It can be noted that both the far-range (mostly −50 dB with −43 dB in few places, where cardboard mask had gaps) and even the near-range (−16 dB) sidelobes were reduced when the masking aperture was used. Beam width along *x*-axis is 0.84 mm and 0.84 mm along y axis.

### 2.2. Procedure of the Integrity Inspection

In integrity inspection experiments, the focused transducer was excited by transformer push-pull pulser SE-TX02-02 (Kaunas University of Technology, Kaunas, Lithuania) [34] using bipolar 60 Vpp rectangular spread spectrum signal (0.7–1.1 MHz range, 40 µs chirp). 

The integrity inspection involved two types of the air-coupled probes. One was a wideband 0.65 MHz center frequency air-coupled probe (Figure 7a) used in resonant ultrasound spectroscopy studies [7]. Probe have been developed by the Ultrasonic and Sensors Technologies Department, CSIC (Madrid, Spain) [5,35]. Probe was made using 20 mm diameter piezoelectric composite and a stack of resonant and nonresonant matching layers [35] to match the transducer acoustic impedance to the air. In such a way, both sensitivity and bandwidth are optimized (bandwidth 90% at −20 dB in thru-transmission mode). Another tested probe (Figure 7b) was rectangular (0.32 MHz center frequency single element [36]). It has been manufactured using the 65 μm polypropylene film HS-03-20BRAL1 (Emfit, Vaajakoski, Finland) glued using the cyanoacrylate glue on 40 mm × 40 mm FR4 printed circuit board (PCB). Rectangular 20 mm × 20 mm copper area on PCB served as bottom “hot” electrode for polypropylene. Aluminum coating of polypropylene served as top ground electrode and was connected to the opposite side of PCB using the conductive copper foil (WE 3003310A 40 µm, Würth Elektronik, Waldenburg, Germany). BNC connector was soldered on the opposite side of PCB.

Probe under inspections was placed in front of the focused transducer at the focal distance (30 mm). 

Received signal was picked up by 1 kΩ input impedance, adjustable gain (14–60 dB) preamplifier PA1008 (AUT Solutions, Fulshear, TX, USA). Preamplifier was followed by low pass 3 MHz bandwidth third-order Butterworth filter. An acquisition system of our own design [37] was used for data collection. Waveforms were sampled using 10-bit ADC operating at 100 MHz. Sampled signals were transferred to a host PC using a high-speed USB 2.0 interface.

System also has *x*, *y*, and *z* coordinates scanner with 10 µm resolution along *x* and *y* axes and 5 µm resolution along *z* axis. The scanning area was either 40 × 40 mm^2^ or 20 × 20 mm^2^ depending on the task, and 0.25 mm step was used. 

The digitized waveforms were cross-correlated with the reference waveform. The reference waveform was selected from original set, by locating the waveform with highest energy. This waveform was filtered by the 4-th order Butterworth bandpass filter with 0.7–1.1 MHz (0.9 MHz center) range. Maximum of the cross-correlation function was used to plot the sensitivity profiles along *x* and *y* axis or 2D C-scan. Slice position was chosen to go across the defect area. Optionally, part of the resulting data set was masked by a round (first probe) or rectangular (second probe) window, corresponding to the region of interest (ROI). ROI corresponded to the active area of the probe investigated (19 mm diameter for the first probe and 22 mm × 22 mm rectangle for the second probe).

## 3. Results

A first set of experiments was carried out using the unmodified focused transducer in order to illustrate the attainable contrast and resolution when no mask is used on the transmitting transducer.

### 3.1. Imaging without Aperture

Results for the round 0.65 MHz probe sensitivity profile inspection are presented in Figure 8. No defects were expected for this probe, so the goal of the inspection was just to confirm that sensitivity is homogeneous over the probe’s surface.

Note the cross-hair marker in C-scan image, which is used to indicate the position of the slice used to provide the amplitude profiles along *x* and *y* axes. Position used here corresponds to the soldering point of the “hot” electrode. Additional mass should alter the reception sensitivity of the probe here. The dark spot (Figure 8c) and the dip (−13.47 dB in Figure 8a,b) in sensitivity profile roughly correspond to this natural defect location.

Artificial defect was simulated by adding the steel spacer (external diameter of 9.85 mm, hole of 4.38 mm, and thickness of 0.8 mm) on the probes’ surface (Figure 9).

Results for the sensitivity profile obtained in case of spacer used as an artificial defect are presented in Figure 10.

The slice position (cross-hair marker in C-scan image) used here corresponds to the center of the spacer. While it was expected that acoustic impedance mismatch will block the reception, the dark ring (Figure 10c) and the dips (−13.47 dB in Figure 10a,b) in sensitivity profile do not correspond to the spacer geometry.

Another artificial defect was simulated, by adding the sticky copper tape (40 µm thick) triangle on the probe’s surface (Figure 11).

The sensitivity profile images obtained when copper tape triangle was used as artificial defect are presented in Figure 12.

Position of the slice (cross-hair marker in C-scan image) corresponds to the center of the copper triangle. It was expected that acoustic impedance will mismatch and copper plus sticky adhesive attenuation will block the reception. However, only edges of the triangle can be distinguished on the image (Figure 12c).

The second, rectangular 0.32 MHz probe, contained a defect (Figure 13) on the top electrode. Surface was accidently rubbed during the assembly process so part of the aluminum coating was scratched away.

In some places, damage was so severe that even the surface of the polypropylene film (white) below the aluminum coating (silver/gray) was clearly visible. While probe size was 40 mm × 40 mm, only 20 mm × 20 mm area was supposed to be sensitive to the acoustic impact because the “hot” electrode was only this size. The sensitivity profiles obtained are presented in Figure 14.

Position of the slice (cross-hair marker in C-scan image) corresponds to the center of electrode damage. It was expected that electrode loss will not provide the reception signal in this area. However, the sidelobes of the focused transducer are so large that sensitivity was reported even outside of the sensitive electrode area (Figure 14c). Defect was not visible, and signal intensity variation was large and present everywhere.

It can be concluded that strict requirements should be applied on focused transducer’s sidelobes in case of sensitivity inspection. The mainlobe area is much smaller (0.7 mm^2^) than the area of the inspected probe (314 mm^2^ for round probe and 400 mm^2^ for rectangular probe), which is hundred times larger. Both mainlobe signals and sidelobes’ signals sum up at the receiving surface, producing either constructive or destructive interference. This ends up in significant and rapid variation of the reported sensitivity. In addition, image contrast is reduced, so small defects cannot be identified.

### 3.2. Tests with Masking Aperture

Application of the masking aperture in the focal spot area of the beam significantly reduces the sidelobes, i.e., far-range sidelobes go down (−50 dB instead of −18 dB for unmodified case). This can be seen from directivity profiles, presented in Figure 4 and Figure 6. Integrity inspection results obtained using the masking aperture are presented below.

Results for the round 0.65 MHz probe sensitivity profile inspection when sidelobes-masking aperture was used are presented in Figure 15.

As already mentioned, no defects were expected for this probe, but soldering point was expected to give some sensitivity reduction. Sensitivity variation in defect-free zones is negligible (a few decibel). It can be concluded that use of the sidelobes-masking aperture provides clear image of the approximately 15 dB sensitivity loss at exact soldering point location. It is interesting to point out that sensitivity loss was due to resonant frequency shift. Additional mass of the solder on the “hot” electrode shifted the resonant frequency, i.e., it became 0.456 MHz so second harmonic (0.9136 MHz) moved into reception frequencies window (0.9 MHz center). Refer Figure 16 for spectral distribution plots along *x* and *y* axes at soldering point location. 

Excitation signal was 0.7–1.1 MHz chirp, this is the reason that no spectral data is available beyond this range. Note the spectral dips at natural defect location.

Results for the sensitivity profile obtained in case of a steep spacer used as an artificial defect are presented in Figure 17.

Both size and position of the artificial defect (refer to Figure 9 for defect photo) introduced are clearly visible. The contrast and the overall image quality are much better than in case of no masking aperture (Figure 10). The spectral distribution plots along *x* and *y* axes across the artificial defect location (Figure 18) indicate that sensitivity at all frequencies have been lost.

Sensitivity profile plots in case of the sticky copper tape triangle as artificial defect are presented in Figure 19.

Defect was slightly raised from the surface (in these experiments, we did not want to damage the surface so tape was not completely glued to the surface), therefore, left edge looks bent. Otherwise, size and position of the artificial defect (refer to Figure 11 for defect photo) introduced are clearly visible, contrast and the image quality are much better than in case of no masking aperture (Figure 12). Spectral distribution plots along *x* and *y* axes across the artificial defect location (Figure 20) indicate that all frequencies have been blocked.

Sensitivity plots in case of the masking aperture use for the second, rectangular 0.32 MHz probe, which contained a defect on the top electrode, are presented in Figure 21.

Sensitivity variation in defect-free zones is negligible (a few decibel). It can be concluded that use of the sidelobes-masking aperture provides clear image of the approximately 25 dB sensitivity loss at electrode loss location.

## 4. Discussion

Inspection of the local sensitivity of air-coupled probes when probing (focused) transducer and inspected (flat beam) probe center frequencies are close creates a challenge. The area of the transmitting element of the focused transducer must be large in order to attain narrow focal spot. Air-coupled operation demands sensitivity, so composite transducer has to be used. Lens focusing is not available, so concave transducer surface has to be constructed using small pieces of piezoelements. The area of the mainlobe is much smaller than the total receiving area of the inspected probe (0.7 mm^2^ vs. 400 mm^2^ in the case analyzed here). On top of this, area of the transmitting transducer is large. Even relatively low signals of the sidelobes accumulate at the receiving surface even if not all are coherent. Furthermore, small pieces of the concave piezoelement (refer Figure 2) produce nonuniform field, which results in increased sidelobes because operation is only in transmission. Therefore, background level in the image is high. This phenomenon creates the impression that probe is sensitive outside its surface (refer Figure 22).

Situation is worsened when relatively large piezoelement pieces are used to produce the spherical surface, required for beam focusing [21]. Application of mechanical structures to alter the transducer directivity is not new. Fresnel zone lenses in front of transducer were proposed in [30,38], additional parabolic reflecting horn was used in [12] and cone-shaped waveguide was applied in [13]. Application of the parabolic reflecting horn was intended for mainlobe width reduction, but the sidelobes remained high (−10 dB according to Figure 6 in [12]). The cone-shaped waveguide is improving the lateral resolution, but near-range sidelobes remain (refer Figure 6 in [13]). Moreover, such waveguide introduces additional delays because of the multimodal propagation and, therefore, pulse duration is expanded. Approach proposed here is different, i.e., masking aperture is placed at the focal spot. Absence of the waveguide reflections does not reduce the temporal resolution.

Far-range sidelobes were suppressed significantly by use of the proposed aperture, i.e., sidelobes went down by 25 dB (from −18 to −43 dB). Moreover, we assume that −43 dB is the artefact, caused by not perfect aperture, i.e., there might have been a leak at outer aperture edges. This assumption can be confirmed by −43 dB sidelobe location (Figure 6a,c)—it is located at the outer edge of the aperture (12 mm from center, which corresponds to the half of aperture diameter). Additional confirmation of this assumption comes from the fact that elsewhere sidelobes are absent, i.e., they are at −50 dB level, defined by the noise floor. Further noise reduction is complicated because hydrophone sensitivity is very low, so averaging takes significant experiment time. 

Even 25 dB sidelobe reduction was sufficient to obtain significant contrast improvement. Signal drop caused by the copper foil defect increased by 20 dB, i.e., from 10 dB in case of unmodified transducer (Figure 12) to 30 dB when masking aperture was applied (Figure 19).

Near-range sidelobes are defined by the shape of the apodization function [39,40]. Directivity is close to *sinc* function in case of uniform apodization. Nonuniform apodization helps to reduce the sidelobes at the price of focal spot broadening [41]. The aperture proposed here can be regarded as equivalent to apodization. Near-range sidelobes’ reduction was not aimed in this application (far-range was targeted). Still, near-range sidelobes were reduced from 6–9 to 10–17 dB. It can be assumed that reason for insignificant reduction was almost trapezoidal apodization function (4 mm rectangular aperture of cardboard plus 2.7 mm blurred aperture of felt). It is worth pointing out that mainlobe width was reduced after aperture application, i.e., unmodified beam was 0.9–1 mm wide and the aperture produced 0.84 mm beamwidth. 

## 5. Conclusions

Experimental results demonstrate that inspection of the air-coupled probes is possible using a focused transducer with a similar frequency. The use of similar frequency relaxes the requirements for focused transducer. Broadband acquisition equipment is not required. Thickness of the matching layers is similar for both transducers, optimal matching to the air is preserved and efficient ultrasound transmission is maintained over the whole propagation path. Demand for similar frequency creates a challenge. Large diameter focused transducer must be used in order to attain a small focal spot. Composite piezoelement must be used to compensate air-matching losses. Concave transducer surface has to be constructed using small pieces of piezoelements because lens focusing is not applicable. Then the area of the mainlobe is much smaller than the total receiving area of the inspected probe or transmitting transducer. Therefore, even relatively low and incoherent signals of the sidelobes accumulate at the receiving surface. Sidelobes-masking aperture was proposed to solve the problem. Aperture is applied in close vicinity to focal spot. It is constructed using a cardboard and felt. Far-range sidelobes were highly suppressed down to noise floor (−50 dB). Even high near-range sidelobes (induced by segments-separating slit in piezoelement) were reduced down to −17 dB. Mainlobe width was reduced from 0.9–1 to 0.84 mm after aperture application.

With such modifications, convenient inspection of the probe integrity can be carried out at significantly enhanced contrast and lateral resolution. Experimental results demonstrate that transduction homogeneity can be inspected for different types of probes with different center frequencies (ranging from 0.3 to 0.65 MHz), using a focused transducer with a similar center frequency (0.9 MHz). Natural and artificial defects of tested probes surface and structure can be detected by the use of proposed method. For instance, the signal drop caused by the copper foil defect improved by 20 dB when masking aperture was applied.

## Figures and Tables

**Figure 1 sensors-20-07196-f001:**
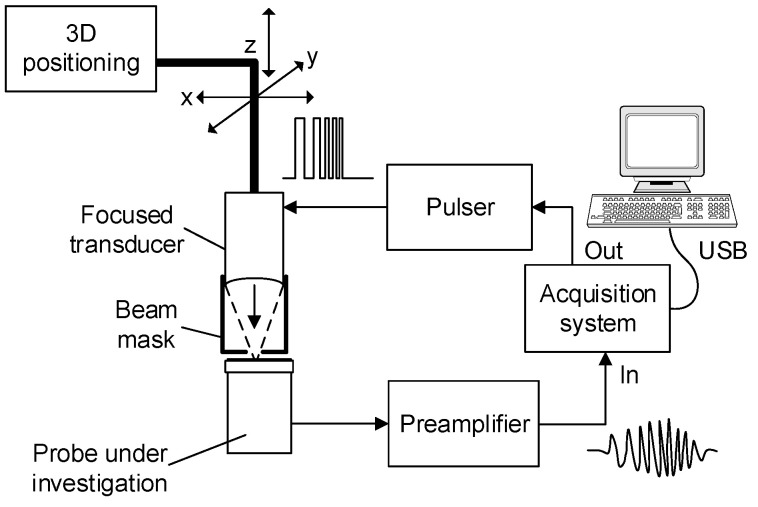
Probe integrity inspection setup.

**Figure 2 sensors-20-07196-f002:**
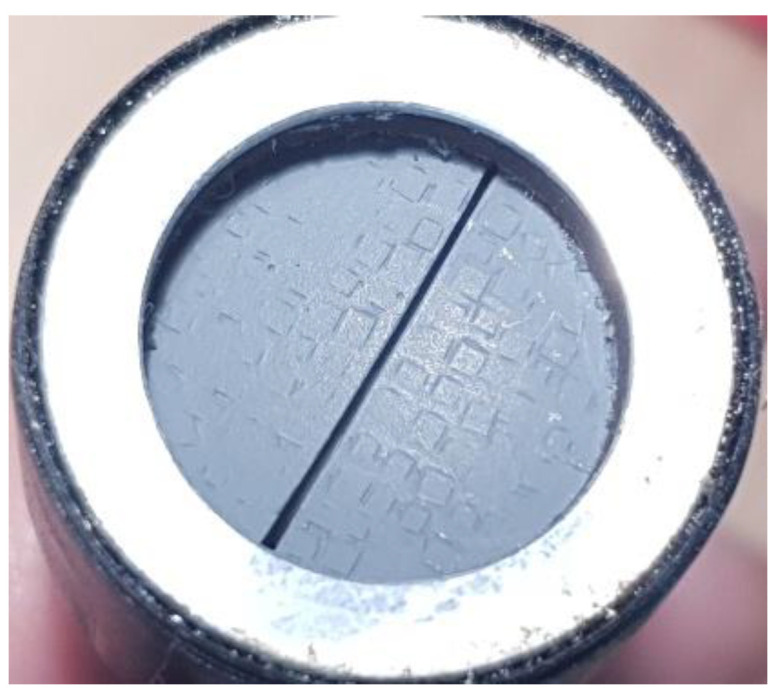
Focused 0.9 MHz 15 mm diameter dual piezoelement transducer used for transmission.

**Figure 3 sensors-20-07196-f003:**
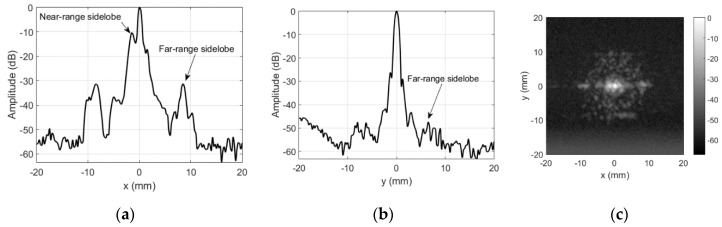
Reflection mode directivity profile of the unmodified focused transducer: (**a**) along *x*-axis at *y* = 0 mm, (**b**) along *y*-axis at *x* = 0 mm, and (**c**) 2D gray-scale profile in dB.

**Figure 4 sensors-20-07196-f004:**
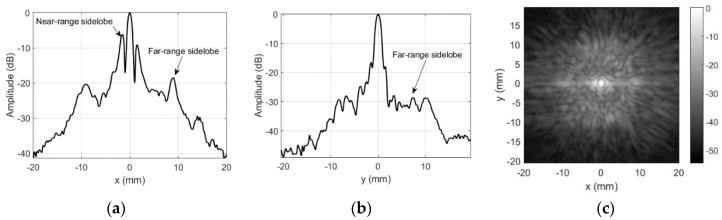
Transmission mode directivity profile of the unmodified focused transducer: (**a**) along *x*-axis at *y* = 0 mm, (**b**) along *y*-axis at *x* = 0 mm, and (**c**) 2D gray-scale profile in dB.

**Figure 5 sensors-20-07196-f005:**
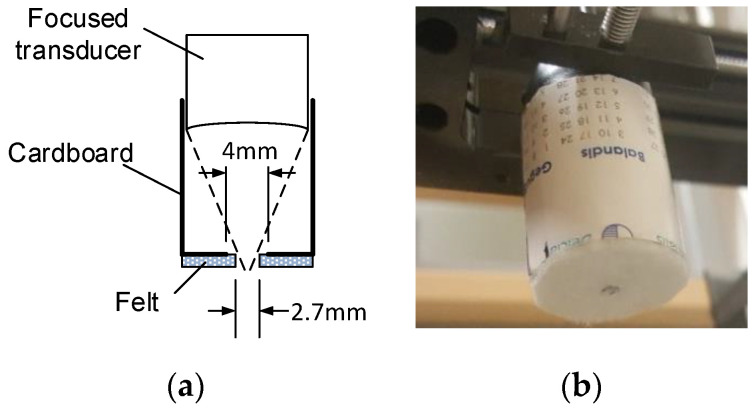
Transducer mask used for sidelobes’ cancelation: (**a**) construction drawing and (**b**) photo of the mask mounted on the focused transducer.

**Figure 6 sensors-20-07196-f006:**
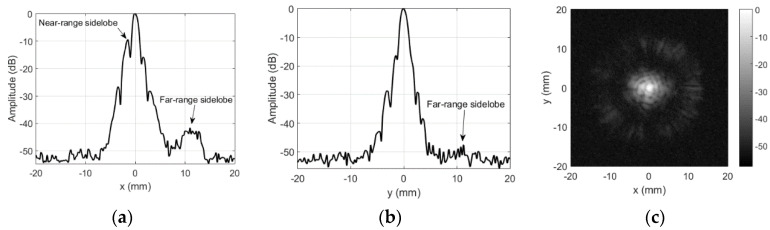
Aperture influence on transmission mode directivity profile: (**a**) along *x*-axis at *y* = 0 mm, (**b**) along *y*-axis at *x* = 0 mm, and (**c**) 2D gray-scale profile in dB.

**Figure 7 sensors-20-07196-f007:**
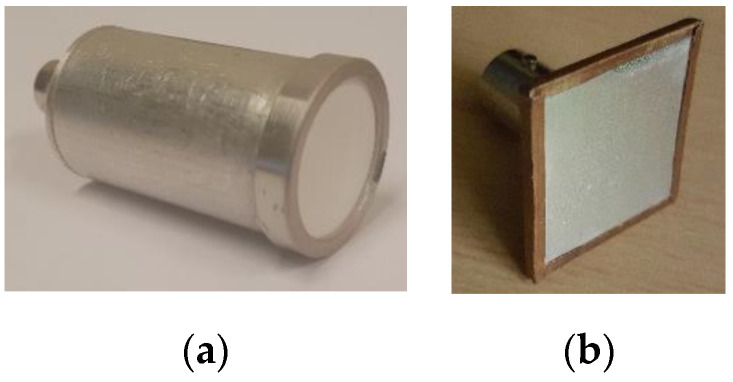
Air-coupled probes used in experiments for integrity inspection: (**a**) round wideband 0.65 MHz probe and (**b**) rectangular single element polypropylene-based 0.32 MHz probe.

**Figure 8 sensors-20-07196-f008:**
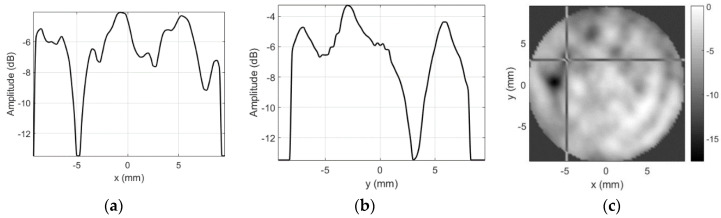
Round probe (no artificial defects) integrity test results using unmodified focused transducer: (**a**) along *x*-axis at *y* = 3 mm, (**b**) along *y*-axis *x* = −4.75 mm, and (**c**) C-scan profile in dB.

**Figure 9 sensors-20-07196-f009:**
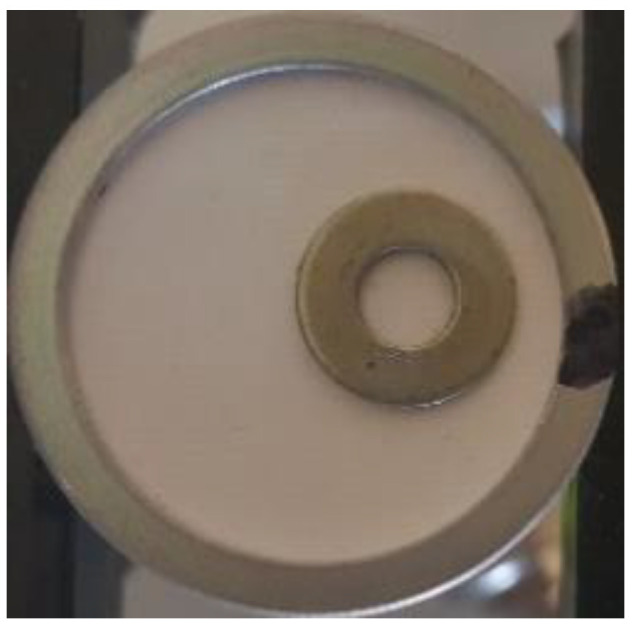
Steel spacer as artificial defect placed on probe’s surface.

**Figure 10 sensors-20-07196-f010:**
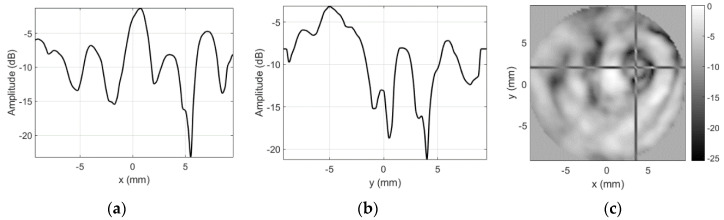
Round probe (steel spacer as artificial defect) integrity test results when using unmodified focused transducer: (**a**) along *x*-axis at *y* = 2 mm, (**b**) along *y*-axis at *x* = 3.5 mm, and (**c**) C-scan profile in dB.

**Figure 11 sensors-20-07196-f011:**
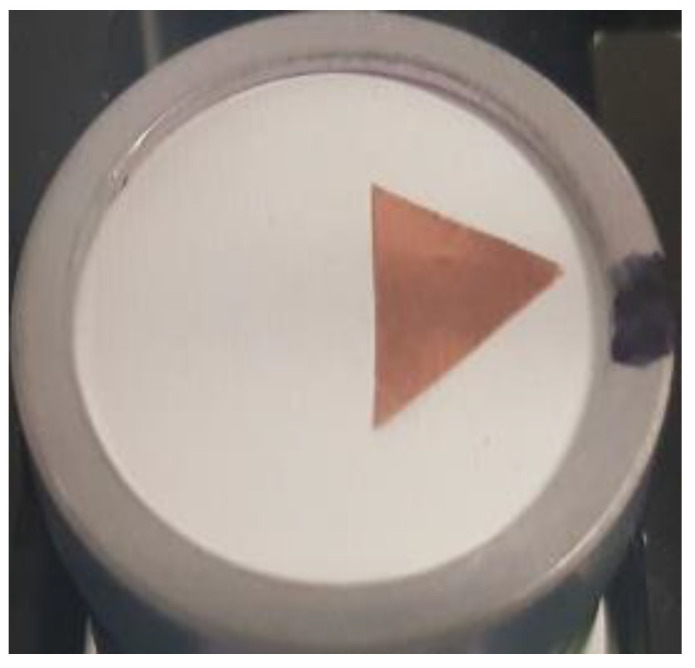
Copper tape triangle as artificial defect placed on probe’s surface.

**Figure 12 sensors-20-07196-f012:**
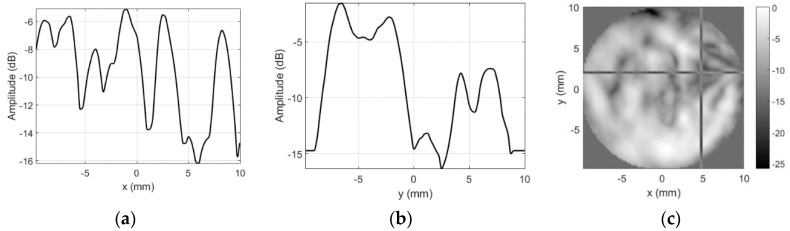
Round probe (copper tape triangle as artificial defect) test results with unmodified focused transducer: (**a**) along *x*-axis at *y* = 2 mm, (**b**) along *y*-axis at *x* = 4.75 mm, and (**c**) C-scan profile in dB.

**Figure 13 sensors-20-07196-f013:**
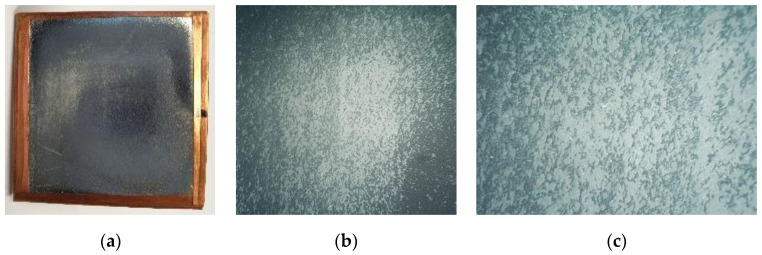
Electrode defect image of the rectangular polypropylene-based probe: (**a**) front view of the whole probe, (**b**) microscope zoom-in over damaged area under 6× magnification, and (**c**) microscope zoom-in over damaged area under 24× magnification.

**Figure 14 sensors-20-07196-f014:**
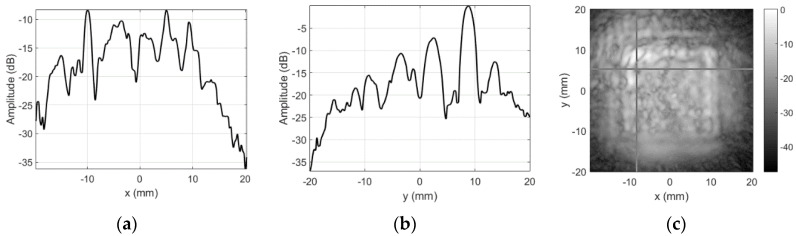
Rectangular probe (no artificial defects) integrity test results when using unmodified focused transducer: (**a**) along *x*-axis at *y* = 5.25 mm, (**b**) along *y*-axis at *x* = −8.25 mm, (**c**) 2D gray-scale profile in dB.

**Figure 15 sensors-20-07196-f015:**
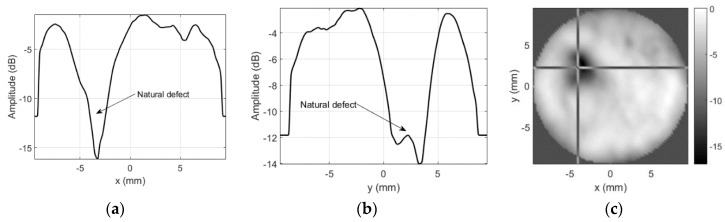
Round probe (no artificial defects) integrity test results using sidelobes-masking aperture: (**a**) along *x*-axis at *y* = 2.25 mm, (**b**) along *y*-axis *x* = −4.25 mm, and (**c**) 2D gray-scale profile in dB.

**Figure 16 sensors-20-07196-f016:**
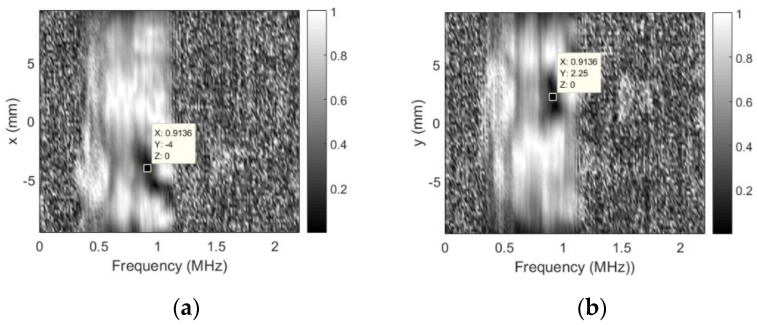
Spectral distribution plots at soldering point location when using sidelobes-masking aperture: (**a**) along *x*-axis at *y* = 2.25 mm and (**b**) along *y*-axis at *x* = −4 mm.

**Figure 17 sensors-20-07196-f017:**
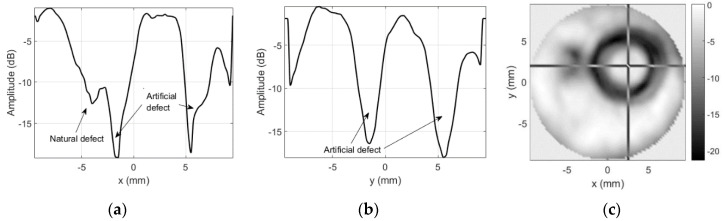
Round probe (steel spacer as artificial defect) integrity test results when sidelobes-masking aperture was used: (**a**) along *x*-axis at *y* = 2 mm, (**b**) along *y*-axis at *x* = 2.5 mm, and (**c**) C-scan profile in dB.

**Figure 18 sensors-20-07196-f018:**
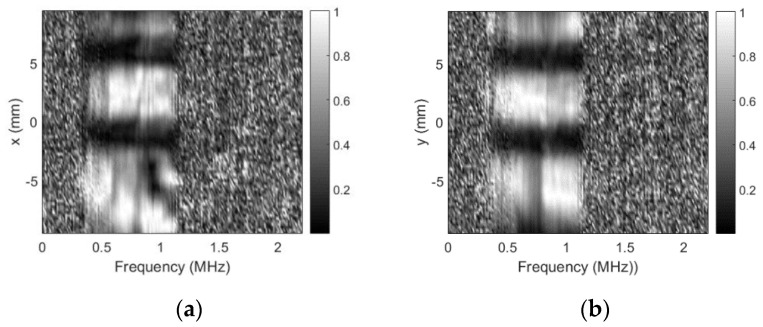
Spectral scans (steel spacer as artificial defect) when using sidelobes-masking aperture: (**a**) along *x*-axis at *y* = 2 mm and (**b**) along *y*-axis at *x* = 2.5 mm.

**Figure 19 sensors-20-07196-f019:**
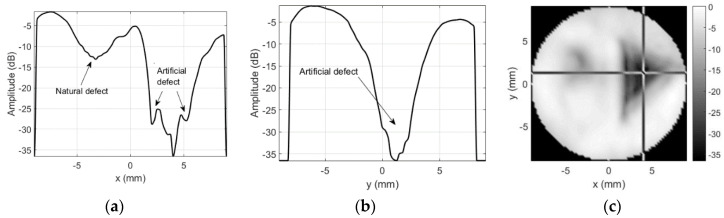
Round probe’s (copper tape triangle as artificial defect) integrity test results when using masking aperture: (**a**) along *x*-axis at *y* = 1.25 mm, (**b**) along *y*-axis at *x* = 4 mm, and (**c**) C-scan profile in dB.

**Figure 20 sensors-20-07196-f020:**
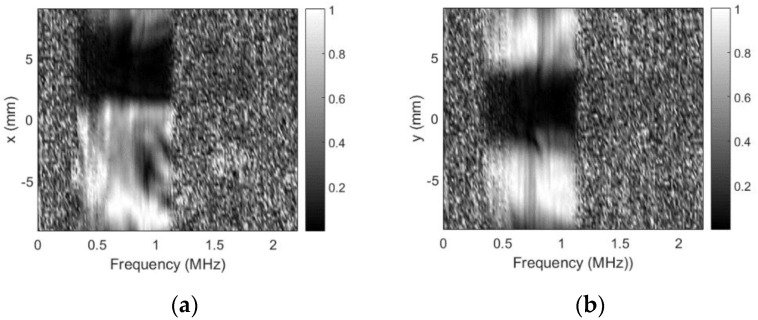
Spectral scans (copper tape triangle as artificial defect) when using masking aperture: (**a**) along *x*-axis at *y* = 1.25 mm and (**b**) along *y*-axis at *x* = 4 mm.

**Figure 21 sensors-20-07196-f021:**
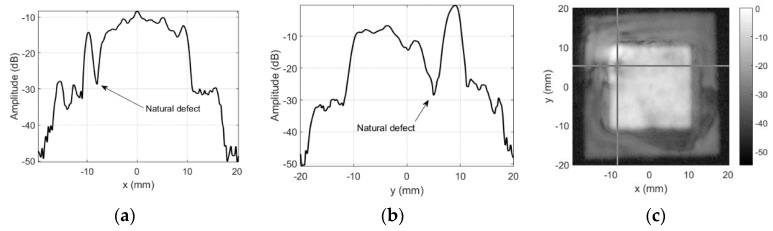
Rectangular probe integrity test results using masking aperture: (**a**) along *x*-axis at *y* = 5.25 mm, (**b**) along *y*-axis at *x* = −8.25 mm, and (**c**) C-scan profile in dB.

**Figure 22 sensors-20-07196-f022:**
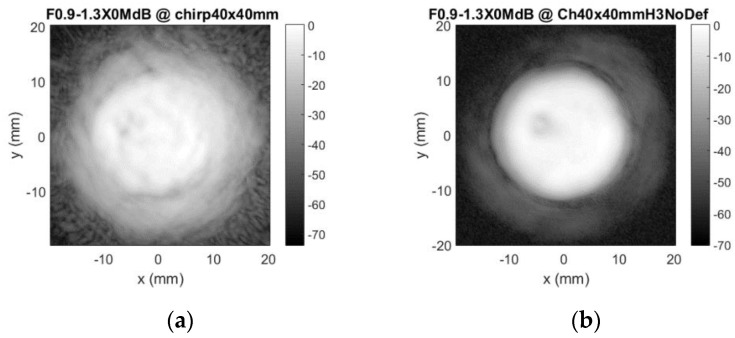
Accumulation of the sidelobe’s signals creates a halo effect if no sidelobe-masking aperture is used (**a**). Application of the aperture at focal spot location can reduce the halo effect (**b**).

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
