# Peer review of "Air-Coupled Ultrasonic Probe Integrity Test Using a Focused Transducer with Similar Frequency and Limited Aperture for Contrast Enhancement"

_sensors, 2020, doi:10.3390/s20247196_

Round 1

Reviewer 1 Report

Dear editor:

Thank you for inviting me to evaluate the article titled “Air-Coupled Ultrasonic Probe Integrity Test by Using a Focused Transducer with Similar Frequency and Limited Aperture for Contrast Enhancement”. In this paper, The authors tested the air - coupled probe can be detected by focusing transducer with similar frequency.The authors have done an excellent technical job. This is definitely worthy of consideration. The article can be accepted.

Author Response

Reviewer 1:

Thank you for inviting me to evaluate the article titled “Air-Coupled Ultrasonic Probe Integrity Test by Using a Focused Transducer with Similar Frequency and Limited Aperture for Contrast Enhancement”. In this paper, The authors tested the air - coupled probe can be detected by focusing transducer with similar frequency.The authors have done an excellent technical job. This is definitely worthy of consideration. The article can be accepted.

Comment: Thank you for your time and appreciation.

Reviewer 2 Report

This manuscript describes a technique for the integrity inspection of an air-coupled ultrasonic probe through surface scanning with a focused beam of another transducer having a similar center frequency. The proposed technique is simple and inexpensive in comparison to other methods requiring reflecting coating and dedicated measurement equipment. The proposed technique also implies an aperture mask realized as simple, low cost hood mounted on the probing transducer.

This manuscript is overall well written, however, it requires general improvement. Followings are the main concerns which should be addressed accordingly.

(1) The introduction section is very brief and the main novelty of the present study in comparison to the preceding works should be further discussed in this section.

(2) Section 2 titled as “Material and methods” is very comprehensive and can be readily subdivided into two parts for better understanding of readers. The first part can describe the measurement setup, probe structure, transducer masking, and initial directivity profiles. The second part can discuss the detailed procedure of the integrity inspection.

(3) The focused transducer shown in Fig. 2 should be properly labeled to illustrate its components as well as dimensions of the slit between two piezocomposite surfaces.

(4) Figure 3 illustrates the reflection mode directivity profiles of the focused transducer. What is the significance of reflection mode directivity profiles in relation to the integrity inspection?

(5) In section 3, the sub-section stated as “Test with masking aperture” should be numbered as 3.2.

(6) Section 4 and 5 are very brief, a detailed discussion regarding the efficacy of aperture placement on the focal spot in comparison to the preceding works should be provided. Similarly the conclusion section should also be a little comprehensive to reflect the significance of the present work in comparison to the precedent.

Author Response

Reviewer 2:

This manuscript describes a technique for the integrity inspection of an air-coupled ultrasonic probe through surface scanning with a focused beam of another transducer having a similar center frequency. The proposed technique is simple and inexpensive in comparison to other methods requiring reflecting coating and dedicated measurement equipment. The proposed technique also implies an aperture mask realized as simple, low cost hood mounted on the probing transducer.

This manuscript is overall well written, however, it requires general improvement. Followings are the main concerns which should be addressed accordingly.

(1) The introduction section is very brief and the main novelty of the present study in comparison to the preceding works should be further discussed in this section.

Comment: Introduction section has been revised.

Reviewer 2:

(2) Section 2 titled as “Material and methods” is very comprehensive and can be readily subdivided into two parts for better understanding of readers. The first part can describe the measurement setup, probe structure, transducer masking, and initial directivity profiles. The second part can discuss the detailed procedure of the integrity inspection.

Comment: We agree with reviewer, section has been split into two parts.

Reviewer 2:

(3) The focused transducer shown in Fig. 2 should be properly labeled to illustrate its components as well as dimensions of the slit between two piezocomposite surfaces.

Comment: Dimensions of the slit and transducer details have been added to manuscript text and figure caption.

Reviewer 2:

(4) Figure 3 illustrates the reflection mode directivity profiles of the focused transducer. What is the significance of reflection mode directivity profiles in relation to the integrity inspection?

Comment: Directivity in reflection mode was presented intentionally, because conventional use of focused transducers is in reflection mode. Even if transducer is used in through-transmission mode, the receiving transducer is usually the same (focused), so directivity is as presented in Fig.3. In such sense there is nothing wrong with this transducer: sidelobes are sufficiently low thanks to spatial correlation. Granularity of piezoelement do not create problems for usual applications (see [9] for even worse granularity case). But in our case we are using transducer just in one direction (transmission), therefore significant problems arise: defects cannot be located. That is why we needed the aperture and that is why we did not study the directivity change in reflection mode after aperture application.

Reviewer 2:

(5) In section 3, the sub-section stated as “Test with masking aperture” should be numbered as 3.2.

Comment: Thank you for spotting. Corrected.

Reviewer 2:

(6) Section 4 and 5 are very brief, a detailed discussion regarding the efficacy of aperture placement on the focal spot in comparison to the preceding works should be provided. Similarly the conclusion section should also be a little comprehensive to reflect the significance of the present work in comparison to the precedent.

Comment: We agree with reviewer, sections 4 and 5 were expanded, adding more discussion on improvement obtained and precedent work.

Reviewer 3 Report

The authors have developed a method to test the integrity of air-coupled ultrasonic probe. The results show that the procedure is useful to the inspection of the probes in ultrasound labs. The work is well-written.

Author Response

Reviewer 3:

The authors have developed a method to test the integrity of air-coupled ultrasonic probe. The results show that the procedure is useful to the inspection of the probes in ultrasound labs. The work is well-written.

Comment: Thank you for your time and appreciation.

Reviewer 4 Report

Dear Authors,

The paper entitled “Air-Coupled Ultrasonic Probe Integrity Test by Using a Focused Transducer with Similar Frequency and Limited Aperture for Contrast Enhancement” deals with methods and materials allowing the inspection of an air-coupled ultrasonic probe. Materials and methods are especially based on a focused transducer in the same frequency range characterized by a controlled aperture.

            The paper is well presented with very convincing results. The only downside is maybe the conclusion that deserves to be fleshed out with few perspectives on the proposed work.

            On the other hand, I do not know enough about the field concerned to discriminate at best the novelty and the originality of the envisaged approach.

            On this basis and subject to this last reservation, I consider this paper accepted in the present form.

            Here are two very minor corrections that I noticed:

  • 2, l. 73: “…if figure 2…” à “…in figure 2…”
  • 8, l. 263: “3.1 Test with masking aperture” à “3.2 Test with masking aperture”

            The last point, which I wanted to point out, concerns the bibliography which seems to me too focused on the authors of the article with a ratio of 11/26 papers written by at least one of the authors.

Best regards.

Author Response

Reviewer 4:

Dear Authors,

The paper entitled “Air-Coupled Ultrasonic Probe Integrity Test by Using a Focused Transducer with Similar Frequency and Limited Aperture for Contrast Enhancement” deals with methods and materials allowing the inspection of an air-coupled ultrasonic probe. Materials and methods are especially based on a focused transducer in the same frequency range characterized by a controlled aperture.

The paper is well presented with very convincing results. The only downside is maybe the conclusion that deserves to be fleshed out with few perspectives on the proposed work.

Comment: We agree with reviewer, both section 4 and 5 were expanded, adding more discussion on improvement obtained.

Reviewer 4:

On the other hand, I do not know enough about the field concerned to discriminate at best the novelty and the originality of the envisaged approach.

On this basis and subject to this last reservation, I consider this paper accepted in the present form.

Here are two very minor corrections that I noticed:

2, l. 73: “…if figure 2…” à “…in figure 2…”

8, l. 263: “3.1 Test with masking aperture” à “3.2 Test with masking aperture”

Comment: Thank you for spotting. Corrected.

Reviewer 4:

The last point, which I wanted to point out, concerns the bibliography which seems to me too focused on the authors of the article with a ratio of 11/26 papers written by at least one of the authors.

Best regards.

Comment: Thank you for your note. We understand your concern. 3 references were removed. The rest remain, because we need them for explanation of equipment used or techniques proposed by us.

Reviewer 5 Report

This work “Air-Coupled Ultrasonic Probe Integrity Test by Using a Focused Transducer with Similar Frequency and Limited Aperture for Contrast Enhancement”. In this manuscript, the authors proposed a technique that does not require reflecting coating which is not applicable to air-matched layers nor expensive and complex equipment. Another novelty of the proposed approach is the use of an aperture to mask the beam sidelobes of the probing transducer. Although, the topic, as well as the work, is interesting, however, the following are changes proposed.

  • The abstract of the article should be rewritten to explain it in a better way. The methodology of the research isn’t explained very well. Provide the application of the proposed research at the end of the abstract as well.
  • There are many grammatical errors throughout the paper even in the abstract, which need to be corrected. Try to avoid unnecessary long sentences. It will be better if the manuscript is revised by a native English speaker.
  • The literature review is ambiguous; include some more recent state of the art papers in the Literature review for better understanding. Add the following paper for applications of PZT sensor in introduction section: https://doi.org/10.3390/en13215528

  • The Introduction part of the article must be revised to make it better structured for the readers. Try to explain the previous work related to different aspects of the current research and connect it with the problem statement in the end i.e. identifying the gap and why was this model necessary to develop. An intense revision is required in this section.
  • In the last paragraph of the introduction section, mention the novelty of this paper more clearly. Moreover, mention the applications of this work.
  • If possible, compare the numerical/experimental data with the data already present in the literature for validation in graphical form.
  • The discussion section should be extended to explain all the work carried out in this research and the physics behind this.
  • In the discussion section, the authors should explain their results more, rather than focusing on literature work. If they want to add literature, add in the form of graphical or tabular comparison and validation so that readers can understand better.
  • What will be the effect if polymer piezo elements used such as PVDF for spherical surface? As they can be more flexible and more efficient in such applications.
  • Extend the conclusion section by providing all the critical summary and outcomes of the results. So that readers can understand it in a better way.
  • This paper seems to be more like a report rather than a research article. Kindly reshape the article as per journal standards.

Author Response

Reviewer 5:

This work “Air-Coupled Ultrasonic Probe Integrity Test by Using a Focused Transducer with Similar Frequency and Limited Aperture for Contrast Enhancement”. In this manuscript, the authors proposed a technique that does not require reflecting coating which is not applicable to air-matched layers nor expensive and complex equipment. Another novelty of the proposed approach is the use of an aperture to mask the beam sidelobes of the probing transducer. Although, the topic, as well as the work, is interesting, however, the following are changes proposed.

The abstract of the article should be rewritten to explain it in a better way. The methodology of the research isn’t explained very well. Provide the application of the proposed research at the end of the abstract as well.

Comment: Abstract has been rewritten to put the explanation in a better way.

Reviewer 5:

There are many grammatical errors throughout the paper even in the abstract, which need to be corrected. Try to avoid unnecessary long sentences. It will be better if the manuscript is revised by a native English speaker.

Comment: Whole manuscript has been revised. Many long sentences have been shortened or split.

Reviewer 5:

The literature review is ambiguous; include some more recent state of the art papers in the Literature review for better understanding. Add the following paper for applications of PZT sensor in introduction section: https://doi.org/10.3390/en13215528

Comment: Strange to receive such imperative demand from reviewer. Paper demanded to include is irrelevant. Relevant publications have been included, literature review has been revised both in introduction and discussion part.

Reviewer 5:

The Introduction part of the article must be revised to make it better structured for the readers. Try to explain the previous work related to different aspects of the current research and connect it with the problem statement in the end i.e. identifying the gap and why was this model necessary to develop. An intense revision is required in this section.

Comment: Intense revision of introductory part was done. Previous work analysis was expanded. Problem statement clarified. What do you mean by “model” is unclear: we do not use modeling here.

Reviewer 5:

In the last paragraph of the introduction section, mention the novelty of this paper more clearly. Moreover, mention the applications of this work.

Comment: Introduction has been revised.

Reviewer 5:

If possible, compare the numerical/experimental data with the data already present in the literature for validation in graphical form.

Comment: Comparison with other works is presented in Discussion section.

Reviewer 5:

The discussion section should be extended to explain all the work carried out in this research and the physics behind this.

Comment: Discussion section extended, also explaining the essence of the work and the physics behind.

Reviewer 5:

In the discussion section, the authors should explain their results more, rather than focusing on literature work. If they want to add literature, add in the form of graphical or tabular comparison and validation so that readers can understand better.

Comment: Wider explanation of the results have been presented in the Discussion part. Literature review is presented in Introduction. Here we only compare the idea with mechanical means of beam modification in order to explain the results more as requested. Validation is based on the quality of the images obtained.

Reviewer 5:

What will be the effect if polymer piezo elements used such as PVDF for spherical surface? As they can be more flexible and more efficient in such applications.

Comment: We doubt whether PVDF is the most suitable material for air-coupled focused transducers. Wrinkles are produced during the forming process. Composite is the better choice. Even though, technique (both aperture application for sidelobes reduction and integrity inspection procedure) is applicable.

Reviewer 5:

Extend the conclusion section by providing all the critical summary and outcomes of the results. So that readers can understand it in a better way.

Comment: Thank you, Conclusions section have been revised and extended.

Reviewer 5:

This paper seems to be more like a report rather than a research article. Kindly reshape the article as per journal standards.

Comment: Whole manuscript has been revised.

Round 2

Reviewer 5 Report

The authors have revised the manuscript significantly and the paper is in good shape now. I believe that this paper can be published in the current form.